# Differences in Occupants' Satisfaction and Perceived Productivity in High- and Low-Performance Offices

**Özgür Göçer [1],\*** , **Christhina Candido [1]** , **Leena Thomas [2]** and **Kenan Göçer [3]**

1 Sydney School of Architecture, Design and Planning, The University of Sydney, Sydney, NSW 2006, Australia
2 School of Architecture, Faculty of Design Architecture and Building, University of Technology Sydney, Sydney, NSW 2007, Australia
3 Faculty of Engineering and Architecture, Beykoz University, Istanbul 34805, Turkey
* Correspondence: ozgur.gocer@sydney.edu.au

**Abstract:** This paper reports the results from a dataset comprising 9794 post-occupancy evaluation (POE) surveys from 77 Australian open-plan offices. This paper specifically focuses on a sub-set of 20 offices ($n = 2133$), identified from ranking 10 offices each, with the least ($n = 1063$) and highest ($n = 1070$) satisfaction scores, respectively. The satisfaction scores were evaluated on the basis of seven factors (i.e., building/office aesthetics and quality, thermal comfort and indoor air quality, noise distraction and privacy, personal control, connection to outdoor environment, maintenance and visual comfort, and individual space). Using the POE survey data from 20 offices, regression analyses and two-way ANOVA tests were carried out to understand the differences in occupants' satisfaction and perceived productivity arising from open-plan offices. According to the statistically significant regression analyses results, it was identified that building/office aesthetics and quality ($\beta = 0.55$, $p < 0.001$) and noise distraction and privacy ($\beta = 0.33$, $p < 0.001$) were the two strongest predictors contributing perceived productivity in low-performance offices. Two-way ANOVA test results for the 10 high-performance offices indicate that the perceived productivity was strongly associated with the office's physical configuration, the employees' working experience, and the working hours at that office.

**Keywords:** indoor environmental quality; open-plan offices; user satisfaction; perceived productivity; office design; post-occupancy evaluation

## 1. Introduction

Office spaces with either no internal walls or few internal freestanding or partial-height partitions are commonly referred to as "open-plan", and this layout became the most common typology found in corporate real estate over the last decades [1,2]. This spatial configuration has been found to facilitate a non-hierarchical working environment [3,4] while enabling communication and collaboration [5], and reduced space requirements and operating costs [6]. Despite these benefits, there is a growing body of research and anecdotal evidence from industry pointing to several issues related to poor indoor environmental quality (IEQ) performance and its negative impacts on office workers [7–10]. IEQ has been proven to negatively affect occupants [11,12]. In order to get rid of the problem of measuring productivity caused by varied outputs from office occupiers, most of the researchers have done productivity studies based on self-assessment questionnaires with a subjective rating [13]. A literature search indicates that key findings of research conducted on satisfaction and perceived productivity can be grouped under the following four key topics of investigation:

- Thermal comfort conditions and indoor air quality (IAQ) [7,11,14–18];

- Visual comfort, access to daylight, and view [19–21];
- Noise distraction, lack of privacy, and communication [22–27];
- Office design [28–33].

For the first topic, thermal comfort and IAQ, research reveals that when these two IEQ dimensions are combined, the situation imposes significant pressure on the overall energy consumption of commercial buildings, due to heating, ventilation, and air-conditioning (HVAC) systems used to keep indoors habitable. While some studies document direct linkages of temperature and self-reported performance at work [34–36], there is still disagreement about "ideal" set-points. Some authors suggest lower temperatures to address distraction and complaints [37] while others make the case in favor of warmer indoors, without losses on reported perceived productivity [38,39]. The other dimension that has been studied was "ventilation rate" (i.e., supply of outdoor air) due to its indirect influence on workers' performance, caused by the impact of ventilation on thermal comfort, air quality, and concentration of indoor generated pollutants [40,41]. In addition, another key finding in fully air-conditioned open-plan offices is that occupants prefer to have a higher-degree of control over their immediate surroundings, including but not limited to thermostats, shading devices, and operable windows [27]. Interpersonal differences when it comes to occupants' thermal preferences have also been extensively documented [16,17].

For a second topic of investigation, Veitch [42] indicated that occupants with access to daylight within 5 m report higher satisfaction levels than the others without a window. A study by Heschong [43] shows that access to daylight would positively impact on mental function and attention. Further, the same study found that there is a strong relationship between various characteristics of views and health. However, the glare potential from windows and/or from a large light source was found to be a distracting factor, adversely affecting productivity [19,20]. Occupants indicated that despite the issues of noise and glare caused by operable windows, they would still prefer them for having a direct connection to the outdoors [44]. Furthermore, the literature supports the argument that having access to external views and appropriate daylighting conditions improves the mood of office workers and has a positive impact on their satisfaction [31,45–47].

The third IEQ-related topic, acoustics, is perhaps the one that received the most attention recently. The aim of increasing social interaction and communication has led to a predominant design goal of diminishing the walls used to separate cellular offices. However, there is significant evidence that proves transition to more open office spaces by minimizing spatial boundaries decreased social interaction and face-to-face communication [48,49]. It is observed that those office workers prefer to send e-mails instead of face-to-face collaboration and donning headphones to shut out their colleagues. Hence, without a physical barrier, they become clearly open for informal interaction, which can be detrimental for productivity [26]. In addition, open-plan arrangements reduce privacy and acoustic comfort [24,50–53]. Hongisto et al. [54] denoted that it is not the sound level of noise, but the intelligibility of irrelevant background noise that determines the distracting power of speech. In addition, Lee et al. [25] found that some of the workers who have high noise-sensitivity could be adversely affected by open-plan office arrangement, due to loss of speech privacy.

Finally, the literature search indicated that "office layout" has also been investigated against IEQ. Office design that proposes optimum balance between encouraging positive interactions and reducing negative distractions [55] could play a significant role in mitigating adverse effects of open-plan offices on workers' satisfaction and perceived productivity. Studies following occupants' pre- and post-relocation scenarios demonstrate that the implementation of well-designed office concepts have a significant impact on workers' perceived productivity and health, specifically in long-term situations [31,32,56–58].

When combined, research in open-plan offices has covered much ground on the negative aspects of poor IEQ on people. Although extremely relevant, this type of research does not necessarily support identification of design strategies that are highly rated by occupants or, consequently, assist in gathering evidence around open-plan office designs that actually work. In addition, there is also a knowledge

gap in understanding which IEQ dimensions receive higher scores in premises where low or high perceived productivity results are reported by workers.

Therefore, this paper aims to identify and compare differences found in open-plan offices with low and high scores given by occupants on overall satisfaction and perceived productivity. To this end, this paper acquired a dataset of 9794 post-occupancy evaluation (POE) surveys from 77 Australian open-plan offices to rank offices based on occupants' satisfaction scores on seven variables (building/office aesthetics and quality, thermal comfort and indoor air quality, noise distraction and privacy, personal control, connection to the outdoor environment, maintenance and visual comfort, and individual space) and perceived productivity. As a result, a total of 20 offices ($n$ = 2133) were then investigated in detail, consisting of 10 offices with the lowest scores (low-performance offices) and 10 with the highest scores (high-performance offices). For these 20 selected offices, traditionally investigated key organizational and spatial factors (the type of work, time spent at work area, window proximity, and workplace arrangement) were also analyzed, in order to understand differences in reported perceived productivity arising from open-plan offices.

## 2. Materials and Methods

### 2.1. Dataset

This paper interrogated a dataset of 9794 post-occupancy evaluation (POE) surveys conducted with the BOSSA (building occupant survey system Australia) time-lapse questionnaire in 77 Australian offices ($n$ = 9794), out of which 20 cases studies ($n$ = 2133) were analyzed in detail for this study. As all the offices are located in Australia, there might be some similarities. The BOSSA time-lapse questionnaire is based on 29 core items, asking occupants to rate their satisfaction on spatial comfort, individual space, indoor air quality, thermal comfort, noise distraction and privacy, visual comfort, personal control, and building image on a seven-point scale (dissatisfied–satisfied). In addition, occupants were also asked to rate their perceived productivity, which was based on a self-assessed measure of their overall satisfaction with their work area, perceived health, and overall satisfaction with the building. Basic information about occupants' descriptive characteristics (i.e., gender, age, type of work, time spent at work, work experience) are also included. Candido et al. [58] provide a comprehensive detail on the BOSSA time-lapse questionnaire.

### 2.2. Data Analysis

2.2.1. Factor Analysis with the Use of Dataset ($n$ = 9794)

As depicted in Table 1, the number of dependent variables from the dataset ($n$ = 9794) used here (29 questionnaire items related to IEQ) was reduced to independent variables (seven-factor items). Statistical analyses have indicated that there are several groups of questionnaire items under seven key factors, named here as building/office aesthetics and quality, thermal comfort and indoor air quality, noise distraction and privacy, personal control, connection to the outdoor environment, maintenance and visual comfort, and individual space (Table 1). These key factors were identified to explain 70.6% of the variance in the data structure. Factor analysis was conducted to establish the underlying data structure with Varimax rotation (oblique solution) to find out if the correlation between the factors was zero.

**Table 1.** Component loading of each factor and questionnaire item.

| Factors | Questionnaire Items | Component Loading the Contribution of Each Variable on Principal Components | | | | | | |
|---|---|---|---|---|---|---|---|---|
| | | 1 | 2 | 3 | 4 | 5 | 6 | 7 |
| Building/office aesthetics and quality | Space for breaks | 0.723 | 0.248 | 0.126 | 0.161 | 0.240 | | |
| | Space to collaborate | 0.715 | 0.208 | 0.108 | 0.142 | 0.125 | | 0.156 |
| | Building aesthetics | 0.698 | 0.248 | 0.103 | | 0.173 | 0.371 | |
| | Work area aesthetics | 0.693 | 0.254 | 0.148 | 0.139 | 0.373 | 0.146 | 0.104 |
| | Interaction with colleagues | 0.604 | 0.184 | | | 0.180 | | 0.376 |
| | Personalization of work area | 0.519 | 0.142 | 0.272 | 0.170 | 0.159 | | 0.348 |
| | Comfort of furnishing | 0.518 | 0.266 | 0.185 | | 0.140 | 0.234 | 0.337 |
| Thermal Comfort and Indoor Air Quality | Humidity | 0.196 | 0.808 | 0.103 | | 0.124 | 0.165 | 0.150 |
| | Air movement | 0.258 | 0.791 | 0.127 | 0.130 | 0.156 | 0.144 | 0.144 |
| | Air quality | 0.288 | 0.790 | 0.120 | | 0.149 | 0.210 | 0.109 |
| | Temperature in winter | 0.176 | 0.739 | 0.129 | 0.167 | | | |
| | Temperature in summer | 0.194 | 0.738 | 0.116 | 0.175 | | 0.118 | |
| Noise distraction and Privacy | Sound privacy | 0.134 | 0.123 | 0.830 | 0.267 | | | |
| | Visual privacy | | | 0.824 | 0.174 | 0.112 | | 0.145 |
| | Unwanted interruption | 0.184 | 0.130 | 0.798 | 0.129 | | | 0.205 |
| | Overall noise | 0.176 | 0.214 | 0.749 | | | 0.141 | 0.190 |
| Personal Control | Personal control of air movement | 0.133 | 0.169 | 0.161 | 0.873 | | | |
| | Personal control of cooling & heating | 0.131 | 0.174 | 0.179 | 0.864 | | | |
| | Personal control of artificial lighting | | | 0.145 | 0.824 | 0.113 | 0.138 | |
| | Degree of freedom to adapt | 0.235 | 0.385 | 0.237 | 0.552 | 0.176 | 0.133 | 0.140 |
| Connection to outdoor environment | Access to daylight | 0.179 | 0.130 | | | 0.865 | 0.180 | 0.114 |
| | External view | 0.258 | 0.122 | 0.125 | | 0.849 | 0.121 | |
| | Connection to outdoors | 0.414 | 0.157 | 0.159 | 0.180 | 0.709 | 0.106 | |
| Maintenance & Visual Comfort | Personal control shading | | 0.163 | 0.112 | 0.162 | 0.273 | 0.671 | 0.201 |
| | Cleanliness | 0.489 | 0.242 | 0.143 | | | 0.611 | |
| | Maintenance | 0.551 | 0.282 | 0.118 | | | 0.591 | |
| | Lighting | 0.104 | 0.274 | 0.127 | 0.158 | 0.209 | 0.584 | 0.287 |
| Individual space | Amount of workspace | 0.201 | 0.170 | 0.228 | | 0.102 | 0.141 | 0.775 |
| | Storage space | | | 0.277 | | | 0.121 | 0.762 |

### 2.2.2. Identifying Focus Groups: Ten Lowest and Highest Performing Offices

The factor analysis process identified seven IEQ factors as the underlying structure of the BOSSA time-lapse survey. Factor scores can be assigned to a surveyed office based on these seven factors. Factor scores were computed by averaging the individual questionnaire item scores (mean response has been selected on the seven-point rating scale for this analysis) comprising each factor. Mean responses from those 77 offices became the basis of creating a dataset (*n* = 9794), consisting of scores for the seven IEQ factors. The mean score of the dataset is an average of the 77 offices' scores, giving equal weight to every office included in the dataset. The differences between the baseline and an individual office score can be indicative of whether or not that office is performing better than offices in the dataset. Each questionnaire item receives equal weight. By using mean satisfaction score results on seven key factors, 10 cases with lower and higher satisfaction levels were identified. The mean values of each of the 10 lowest- and highest-performing offices (named here as 10-LPO and 10-HPO, respectively) were compared against the dataset (*n* = 9794) mean score.

POE surveys represented in the 20 case studies were conducted at least six months after occupants occupied the office between the years of 2016 and 2017. Table 2 depicts basic information for all 20 offices, including sample sizes, office layout, tenant, tenant certification, and desk arrangement. Surveys are often conducted due to requirements from the Green Building Council of Australia's (GBCA) green star rating scheme. Therefore, a significant portion of offices investigated here are green star-certified spaces. In addition, these offices are located within a building that holds a valid rating awarded by the National Australian Built Environment Rating System (NABERS), which is typical for commercial office buildings in Australia. The office tenants are from various industry types, including consultancy, non-profit organizations, government, design and consultancy, and tertiary education. Sample sizes varied widely (from 25 to 383 questionnaires), which was expected considering each organization occupied just one or several floors within a building, which is common practice for office buildings in Australia. Offices are mostly consisting of open-plan typology (with and without partitions) and five of them have occupants' seating in a non-fixed location (Table 2).

**Table 2.** Basic information about surveyed offices.

| Focus Group | Ranking | Sample Size N = 2133 | Office Layout | Tenant | Tenant Certification | Workplace Arrangement |
|---|---|---|---|---|---|---|
| 10-LPO | 1 | 82 | Private office & Private office shared with others 8.7%; open-plan with high partitions 24.6%; open-plan with low partitions 61.0%; open-plan without partitions 5.6% | Technology | Green Star | Fixed location |
| | 2 | 78 | | Design & consultancy | - | Fixed location |
| | 3 | 82 | | Consultancy | - | Fixed location |
| | 4 | 28 | | Property industry | Green Star | Fixed location |
| | 5 | 30 | | Design & consultancy | Green Star | Fixed location |
| | 6 | 232 | | Government | - | Fixed location |
| | 7 | 383 | | Finance | Green Star | Non-fixed location |
| | 8 | 26 | | Tertiary education | Green Star | Fixed location |
| | 9 | 25 | | Tertiary education | - | Non-fixed location |
| | 10 | 97 | | Technology | Green Star | Fixed location |
| 10-HPO | 1 | 112 | Private office & Private office shared with others 2.6%; open-plan with high partitions 2.9%; open-plan with low partitions 21.3%; open-plan without partitions 73.2% | Government | - | Fixed location |
| | 2 | 25 | | Property industry | Green Star | Non-fixed location |
| | 3 | 32 | | Design & consultancy | Green Star | Fixed location |
| | 4 | 322 | | Property industry | Green Star | Fixed location |
| | 5 | 57 | | Non-profit | Green Star | Fixed location |
| | 6 | 28 | | Property industry | Green Star | Non-fixed location |
| | 7 | 300 | | Consultancy | Green Star | Non-fixed location |
| | 8 | 51 | | Property industry | Green Star | Fixed location |
| | 9 | 39 | | Property industry | Green Star | Fixed location |
| | 10 | 104 | | Property industry | Green Star | Fixed location |

Of the entire sample, 29.5% of the respondents were female. Only 2.8% of the respondents were over 50 years old. The job category that participated most strongly (39.3%) was professional. The percentage of participants who had been working at the company for more than five years was 20.5%. Further, 73.2% of the respondents were full-time (>30 h) workers. The respondents'

(office workers') profiles are summarized in Table 3. The data were analyzed using statistical methods, with the use of SPSS version 24.

**Table 3.** Summary of occupant profiles in focused groups (10 lowest-performing offices (LPO) and 10 highest-performing offices (HPO)).

| Gender | Female 29.5% | | Male 55.3% | Prefer not to Respond 15.1% | |
|---|---|---|---|---|---|
| Age | Over 50 years old 2.8% | | 31–50 years old 55.4% | 30 years old or under 41.8% | |
| Job category | Administrative 21.4% | Technical 13.7% | Professional 39.3% | Managerial 18.1% | Other 7.5% |
| Working experience at the office | >5 years 20.5% | 2–5 years 17.5% | 1–2 years 11.9% | 7–12 months 23.1% | <6 months 26.9% |
| Working hours in a typical week | >30 h 73.2% | | 11–30 h 20.4% | <10 h or less 6.3% | |

### 2.2.3. Further Analyses with the Focus Groups

Apart from the 29 questionnaire items used in the factor analysis above, the BOSSA time-lapse survey also has the question for perceived productivity. Since the factor analysis extracted seven IEQ factors that are uncorrelated (orthogonal) with each other, the association of seven factors with the four global evaluations can now be examined by regression analysis. Since both dependent and independent variables are numeric, three separate linear regression analyses for dataset and each focus group (10-LPO and10-HPO) were conducted. Linear regression analyses were done for each of factor item taken as the independent variable, where perceived productivity was the dependent variable.

As a useful technique to reveal and determine the relationships between three or more independent variables, two-way ANOVA analyses were carried out for each focus group; 10-LPO and 10-HPO. In this study, the significance level was determined to be 0.05. With the help of ANOVA analysis, the effects between more than one categorical independent variable on the dependent variable could be observed. The analyses were done to investigate the interrelations between the categorical variables of spatial configuration such as office layout, workplace arrangement, being close to a window, and descriptive properties of employees such as age, gender, working experience, type of work, and working hours in a week.

## 3. Results

### 3.1. Identifying High- and Low-Performance Offices

The 10 lowest- and 10 highest-performing offices could be selected by ranking the mean scores for the seven IEQ factors. The mean values of each of the 10 lowest and highest performing offices were compared against the dataset ($n = 9794$) mean score, as seen in Figure 1. In order to compare the offices' mean scores against the dataset ($n = 9794$), a performance index (PI) was used. A positive value indicates the office is performing better, while a negative value indicates the office performance is worse than the mean score of 77 offices (Table 4).

In general, 'Noise distraction and privacy' and 'Personal control' consistently presented the lowest scores, while 'Individual space' was the factor that had the highest score for the dataset ($n = 9794$) and 10-LPO, and 'Maintenance and Visual Comfort' and 'Building/office aesthetics and quality' resulted in high scores for 10-HPO. When the mean scores of 10-HPO and 10-LPO were compared, the biggest gaps were recorded for the two factors 'Connection to the outdoor environment' and 'Building/office aesthetics and quality'.

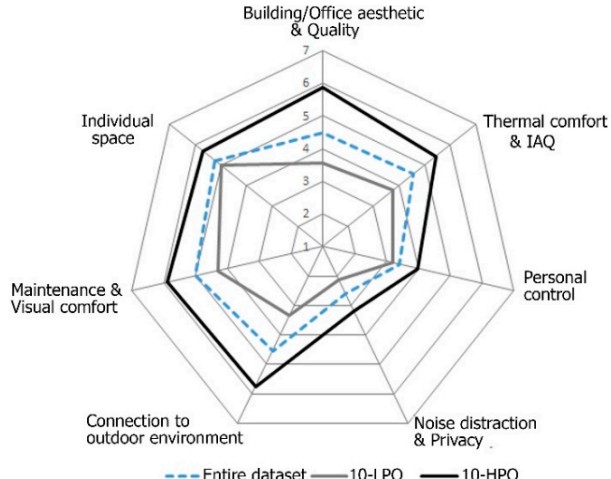

**Figure 1.** Mean scores of each factor for the dataset (*n* = 9794), 10-LPO, and 10-HPO (7 is the highest score and 1 is the lowest).

**Table 4.** Mean scores and performance index (PI) values of 10-LPO and 10-HPO.

| Factors | Mean Scores of Dataset | 10-LPO Mean Scores | PI of 10-LPO | 10-HPO Mean Scores | PI of 10-HPO |
|---|---|---|---|---|---|
| Building/office aesthetics & quality | 4.50 | 3.55 | −0.95 | 5.88 | 1.38 |
| Thermal comfort and Indoor Air quality | 4.56 | 3.75 | −0.80 | 5.43 | 0.87 |
| Personal Control | 3.41 | 3.19 | −0.22 | 4.00 | 0.58 |
| Noise distraction and privacy | 2.58 | 2.12 | −0.46 | 3.18 | 0.59 |
| Connection to the outdoor environment | 4.54 | 3.34 | −1.19 | 5.74 | 1.21 |
| Maintenance & Visual Comfort | 4.99 | 4.29 | −0.70 | 5.90 | 0.91 |
| Individual space | 5.22 | 4.98 | −0.24 | 5.69 | 0.47 |

A detailed evaluation is given below for each focus group:

10-LPO: Out of the seven factors investigated (Figure 1), "Noise distraction and privacy" had the lowest mean score (2.12, Table 4). When the mean scores of the dataset (*n* = 9794) and 10-LPO were compared, the biggest difference was identified for the "Connection to the outdoor environment (–1.19, Figure 1)" that included questionnaire items such as access to daylight, external view, and connection to the outdoor (Table 1).

10-HPO: "Building/office aesthetics and quality" presented the biggest score difference when compared to the dataset (*n* = 9794) (Figure 1). This factor included the following questionnaire items: Space for breaks and collaboration, building aesthetics, interaction with colleagues, personalization of work area, and comfort of furnishings (Table 1). The second biggest score difference when compared to the dataset (*n* = 9794) was found for connections to the outdoor environment (1.21, Table 4).

*3.2. Exploring Predictors Contributing to Perceived Productivity*

The mean scores of the respondents rating of the influence of the work area on their productivity were listed as 4.06, 4.62, and 5.48 for the 10-LPO, dataset (*n* = 9794), and 10-HPO, respectively. In order to understand how the IEQ factors have an impact on these scores, three separate linear regression analyses were conducted for the dataset and each focus group. Perceived productivity item was considered as an independent variable and the seven IEQ factors (i.e., component scores for each of the seven dimensions) as predictors. Table 5 presents the $R^2$s and standardized regression coefficients (β) of the three regression models.

**Table 5.** Results of regression data analyses for 10-LPO and 10-HPO.

| Factors Items (Independent Variables) | Dataset (*n* = 9794) Perceived Productivity | | 10-LPO Perceived Productivity | | 10-HPO Perceived Productivity | |
|---|---|---|---|---|---|---|
| | $R^2$ | β | $R^2$ | β | $R^2$ | β |
| Building/office aesthetics and quality | **0.171 *** | 0.41 | **0.108 *** | 0.32 | 0.115 | 0.33 |
| Thermal comfort & Air quality | 0.076 | 0.27 | 0.068 | 0.26 | 0.077 | 0.27 |
| Personal control | **0.122 *** | 0.34 | **0.072 *** | 0.26 | **0.209 *** | 0.45 |
| Noise distraction & privacy | 0.028 | 0.16 | 0.038 | 0.19 | 0.026 | 0.16 |
| Connection to the outdoor environment | 0.040 | 0.20 | 0.004 | 0.06 | 0.029 | 0.17 |
| Maintenance & Visual Comfort | 0.037 | 0.19 | 0.040 | 0.19 | 0.015 | 0.12 |
| Individual space | 0.049 | 0.22 | 0.035 | 0.18 | **0.251 *** | 0.50 |

* The two strongest predictors were indicated as in bold. Note: *p*-values for all factor items are statistically significant (*p* < 0.001).

'Building/office aesthetics and quality' (for dataset β = 0.41 and for 10-LPO β = 0.32, *p* < 0.001) and 'Personal control' (for dataset β = 0.34 and for 10-LPO β = 0.26, *p* < 0.001) (Table 5) were the two strongest predictors contributing to perceived productivity within 10-LPO. Meanwhile, the two strongest predictors contributing to the perceived productivity of 10-HPO regression results were 'Individual space' (β = 0.50, *p* < 0.001), and 'Personal control' (β = 0.45, *p* < 0.001) (Table 5). The IEQ factor "Individual space" consists of questionnaire items regarding amount of workspace and storage space. "Personal control" IEQ factor consists of the questionnaire items regarding personal control over air movement, cooling and heating, artificial lighting, and degree of freedom to adapt.

### 3.3. Human, Organizational, and Spatial Factors Related to Perceived Productivity

To understand the key drivers for perceived productivity arising out of the open-plan offices surveyed within this research study, focus groups were studied in detail. The relationship between the categorical variables, consisting of human- and organizational-related factors (gender, age, working experience, working hours, working experience, job category, workplace arrangement) and spatial-related factors (office layout, window proximity) and their combined impact on perceived productivity have been investigated by performing two-way ANOVA test for each categorical variable combination. However, no significant relationship was found for the focus group 10-LPO. On the other hand, significant relationships were found for the combination of some variables for the focus group 10-HPO. According to the results (Table 6), it was observed that perceived productivity was mostly related to the occupants' working experience and hours at that office, window proximity, office layout, and workplace arrangement.

**Table 6.** Results of two-way ANOVA tests done for 10-HPO.

| Relationship between | Source | Sum of Squares | df | Mean Square | F | Sig. |
|---|---|---|---|---|---|---|
| Work area close to a window & Office layout | Window | 0.044 | 1 | 0.044 | 0.022 | 0.881 |
| | Office_layout | 14.512 | 4 | 3.628 | 1.842 | 0.118 |
| | Window*Office_layout | 26.359 | 3 | 8.786 | 4.462 | 0.004 |
| Work area close to a window & Working experience | Window | 63.160 | 1 | 63.160 | 31.836 | 0.000 |
| | Time_bldg | 6.994 | 4 | 1.749 | 0.881 | 0.474 |
| | Window*Time_bldg | 27.082 | 4 | 6.770 | 3.413 | 0.009 |
| Office layout & Working hours in a week | Office_layout | 12.001 | 4 | 3.000 | 1.457 | 0.213 |
| | Hours_week | 12.925 | 2 | 6.462 | 3.137 | 0.044 |
| | Office_layout*Hours_week | 32.535 | 8 | 4.067 | 1.974 | 0.047 |

**Table 6.** *Cont.*

| Relationship between | Source | Sum of Squares | df | Mean Square | F | Sig. |
|---|---|---|---|---|---|---|
| Office layout & Working experience | Office_layout | 9.559 | 5 | 1.912 | 0.942 | 0.453 |
| | Time_bldg | 13.229 | 4 | 3.307 | 1.630 | 0.164 |
| | Office_layout*Time_bldg | 61.413 | 16 | 3.838 | 1.891 | 0.018 |
| Office layout & Workspace arrangement | Office_layout | 22.578 | 5 | 4.516 | 2.215 | 0.051 |
| | Workspace_arrangement | 5.553 | 1 | 5.553 | 2.724 | 0.099 |
| | Office_layout * Workspace_arrangement | 20.669 | 4 | 5.167 | 2.535 | 0.039 |

* refers the relationship between the categorical variables and their combined impact on perceived productivity.

The influence of spatial factors (window proximity and office layout) on perceived productivity was further explored. Figure 2 shows the relationship of window proximity to the office layout and working experience. The occupants in private offices have stated a higher perceived productivity for these two options. Considering the fact that 73.2% of office workers are working in open-plan offices without partitions, the influence of these key factors on this office layout gains more importance. Having a workspace close to a window positively affects perceived productivity, especially for the open-plan office with low partitions and without partitions. Duration of employment (i.e., work experience) is seen as another determinant factor when it is interrelated with window proximity. When the employment duration was taken into account, the group working for more than five years has been most adversely affected by the lack of outdoor environment connection.

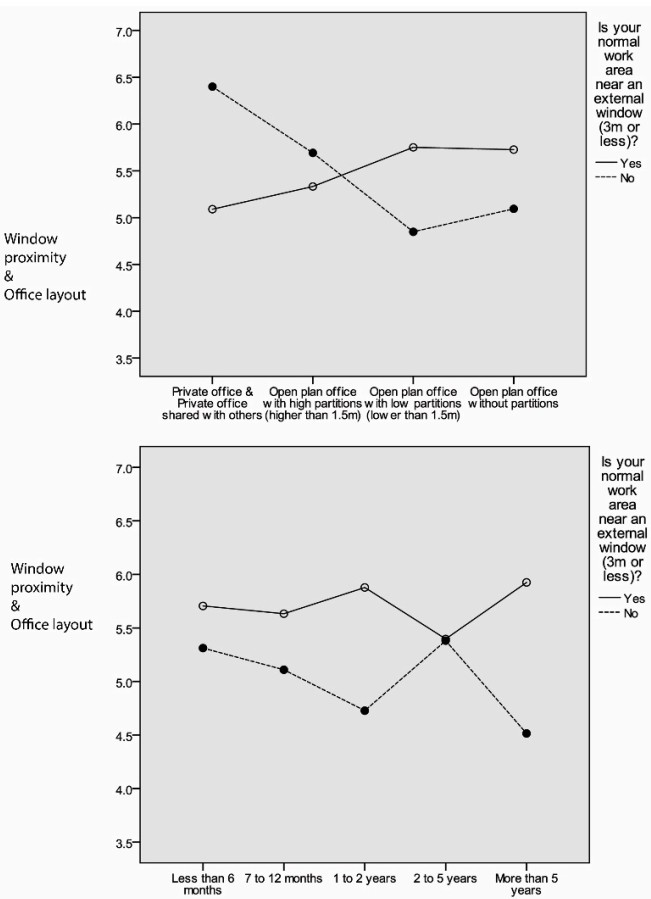

**Figure 2.** Two-way ANOVA results of window proximity influence on perceived productivity by office layout and working experience (4 = neither negative nor positive, 7 = very positive).

The working hours' relationship with the other key factors was also investigated; 73.2% of respondents have been working for more than 30 h in a week. The study with the focus group, 10-HPO, indicates that open-plan office with low partitions is the best type of office layout considering long working hours (Figure 3).

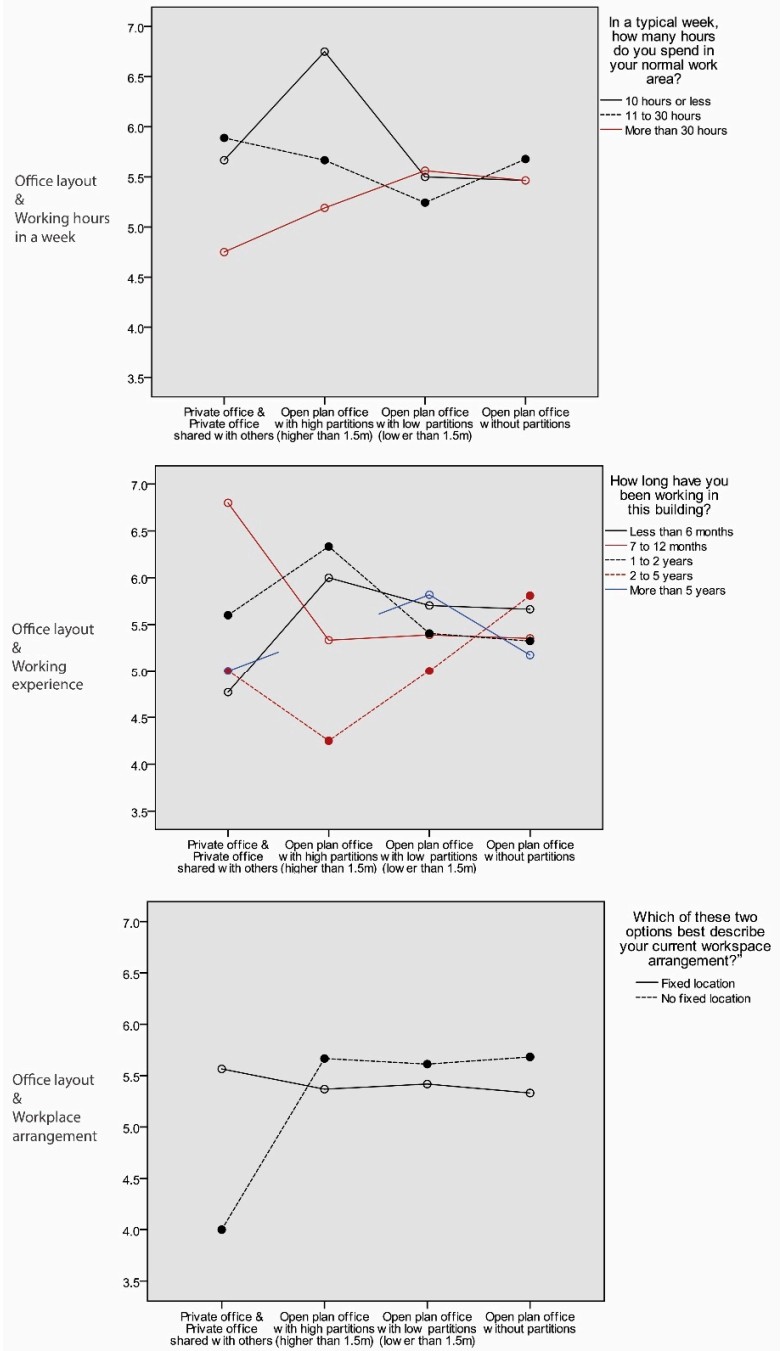

**Figure 3.** Two-way ANOVA results of office layout influence on perceived productivity by working hours and experience and workplace arrangement (4 = neither negative nor positive, 7 = very positive).

Working experience is another determinant factor when it is interrelated with office layout. Although the newcomers (less than one year) have represented high scores at private and shared offices, the level of perceived productivity has dramatically decreased when the working experience increased by two to five years. The open-plan office with low partitions has the best performance

among the other open-plan office types when longer working periods (more than five years) have been taken into account (Figure 3).

The workplace arrangement has been classified in to two group; Fixed location, which includes exclusive and shared use of the same workstation, and non-fixed location, which varies (e.g., activi-based workspace, flexi-desk, etc.) with activities. According to Figure 3, it can be concluded that having a non-fixed location workstation has a positive effect on perceived productivity, except for shared offices.

## 4. Discussion

The results of the analyses based on seven IEQ factors and ranking the 77 offices according to their mean scores of these seven factors are discussed below under the headings of occupants' satisfaction and perceived productivity.

### 4.1. Occupants' Satisfaction

LPO: In accordance with the IEQ factor mean scores for 10-LPO, "Noise distraction and privacy" has the lowest mean score with a significant difference (Figure 1). This finding is compatible with the studies that identified it as the main issue in open-plan offices [24,50–53]. Office workers not only complained about background noise, but also noise coming from the outdoors and HVAC systems.

When the mean scores of 10-LPO and 10-HPO were compared, the difference in "Building/office aesthetic and Quality" and connection to the outdoor environment IEQ factors was very striking. Depending on the complaints of occupants outlined below, it was observed that the spatial-related issues caused a decrease in satisfaction level. The complaints about office design included insufficient private room and/or quiet room for the purposes that require concentration and focusing, limited access to daylight and external view, and overcrowded desks. It was also recorded that, with the lack of an adequate number of meeting rooms for collaborative work, kitchen, relaxation areas, and even workstations were used for meetings. Some quotations from the occupant surveys are given below:

*"Lunchroom and kitchen areas are frequently used as work areas due to lack of collaboration space".*

*" . . . there are not enough desks for officers and what desks there are overcrowded and noisy due to the overcrowding . . . " there is no space to store field equipment which is usually stored under desk and then reduces space for ease of using workspace and can cause back issues and pose as trip hazards".*

*" . . . kitchen area is too small only has two seats. desk size is so small there is nowhere to eat my lunch, also seat opposite a wall and have no view."*

Apart from this, occupants were also complaining about their basic demands for physical requirements such as fresh air and thermal comfort. The following statements have indicated how the occupants provide response to their work environment:

*"Staff on southeastern side have lap rugs and wear their outdoor coats inside, staff on the other side are wearing summer clothes".*

*" . . . people on one side of the building can be boiling while people on the other side are freezing".*

*"$CO_2$ levels are too high and oxygen too low, particularly in the afternoon the air does not seem fresh. This has a significant impact on my alertness and productivity".*

The ratio of complaints on IEQ factor "Thermal comfort and IAQ" was documented as 15.2% and the request–response dissatisfaction rate was very high at 54.7%.

*"The responsiveness of building management to fix air conditioning issues is abhorrent."*

HPO: The analysis shows that, for the users of the studied 10-HPOB, the most pleasant and satisfactory dimension is the "Maintenance and visual comfort" (the mean score is 5.90, Table 4). However according to the PI, which allows us to make a comparative assessment among dataset

and highest- and lowest-performing buildings, "Building/office aesthetics and quality" presented the biggest score difference (Figure 1, Table 4). This factor includes the following questionnaire items: Space for breaks and collaboration, building aesthetics, interaction with colleagues, personalization of work area, and comfort of furnishing (Table 1). The second highest score difference when compared to the entire dataset was found for "Connection to the outdoor environment" (1.21, Table 4), which includes questionnaire items access to daylight, external view, and connection to the outdoors (Table 1). This finding supports the significance of innovative design [28,32], communication and collaboration [26,59], and aesthetics [60] on office worker's satisfaction. With respect to occupants' satisfaction, the study showed significant relationships with access to daylight and view as indicated in other similar studies [45–47].

The satisfaction rate of "Thermal comfort and Indoor Air quality" factor was higher (the mean score is 5.43, Table 4), which includes the following questionnaire items: Temperature in winter and summer, air movement, air quality, and humidity. Only 6.3% of the occupants have indicated that they have complaints on any of the issues such as indoor air temperature, air movement, lighting, and other. People who want temperature warmer was the in the first order in the complaint list with the 25.0%. The request response dissatisfaction rate was 33.3%.

The most unsatisfactory IEQ factor is 'Noise distraction and privacy" (the mean score is 3.18, Table 4). This finding is consistent with the outcomes of relevant studies found in the literature. The comparatively lowest mean score of the IEQ factors points out that the noise level and acoustic problems still exists although the design of the building was satisfactory.

In accordance with the PI, the lowest value is "Individual space", which covers the items as the amount of workspace and storage space. This is a concomitant failure of the design solution in new working spaces by shrinking the overall amount of space dedicated to individual work and increasing collaborative and group spaces [61].

### 4.2. Perceived Productivity

The importance of spatial design was emphasized by many researchers [28–30,32], but it is observed that individual space and IEQ factor "Personal control" are becoming the occupant's priority in a well-designed office. The recent studies on open-plan offices have highlighted the importance of individual space and personal control as having significant and positive impact on perceived productivity [12,21].

The regression analyses in Table 5 have demonstrated that the two strongest predictors were 'Individual space' ($\beta$ = 0.50, $p < 0.001$) and 'Personal control' ($\beta$ = 0.45, $p < 0.001$) for perceived productivity. This reflects that more attention should be paid to solve the problems of the rising voice of inadequate personal storage and work area and adjusting noise level and maintaining privacy.

There is a significant difference between the perceived productivity scores of the occupants who have a connection to an external window with more than five years' experience and the others. These findings coincided with the results of the studies verifying that making visual contact with the outdoors reduces office workers' stress and promotes their quality of life [45,62].

The other determinant factor for perceived productivity was the height of partitions, for 10-HPO especially. Open-plan offices with higher partitions have lower value when compared with other open-plan office types, especially for the occupants working long hours, whereas it is the opposite for occupants working less than 10 hours (Table 6). When it is considered that 73.2% (Table 3) of employees are working long hours, it is evident that a workstation with an open-plan office with low partitions has shown the best performance for 10-HPO.

## 5. Conclusions

Over the last decades, the major issues arising from open-plan offices have been identified and discussed in length in academia and industry. From the inadequacies of indoor environmental quality (IEQ) provided in these offices, to links with lower productivity levels, a considerable body of work has

been devoted to understanding occupants' dissatisfaction in such working environments. This paper has focused on results from perceived productivity observed in high- and low-performance open-plan offices in Australia. Seven factors, namely building/office aesthetics and quality, thermal comfort and indoor air quality, noise distraction and privacy, personal control, connection to the outdoor environment, maintenance and visual comfort, and individual space were included in the analysis. Results from this study suggest links between high perceived productivity levels and office design. The top 10 offices where high levels of satisfaction were reported from occupants presented high satisfaction results for space for breaks and collaboration, building and work aesthetics, interaction with colleagues, personalization of work area, and furnishing comfort. Results also flag the need for more workspace and storage, as well as personal control over systems and adaptability.

In addition, this study also reported results from analysis on key traditionally investigated human (gender, age), organizational (the type of work, type spent at work area, and workplace arrangement) and spatial factors (window proximity and office layout). The spatial features and organizational factors showed links with perceived productivity reported by occupants, especially for the occupants that are working longer hours in a week and having longer experience at that work, open-plan offices with low partitions performed better than the others. If the open-plan office design was supported by non-fixed workspace arrangement, then the influence of workspace on perceived productivity would be higher.

A limitation of this study is its reliance on availability and access to offices, where all were located in Australia. This might have created a familiarity. Since there is not a universally accepted measurement of productivity, a self-assessed measure was used. Similar research has found this as a justifiable consideration [13].

Consequently, the above study also points towards the shortcomings of open-plan offices, including its impact on satisfaction and perceived productivity support. Combined, results from low-performance offices suggest that open-plan offices investigated here could benefit from:

- Improved building and workplace aesthetics,
- Abundant use of zoning, strategically placed to accommodate several work-related activities including, but not limited to quietness, focus, and concentration,
- Abundant use of space for breaks, collaboration, and communication,
- Maximizing occupants' access to daylight and connecting to outdoor environments.

**Author Contributions:** The authors contributed to the paper in the following way: Conceptualization, Ö.G., C.C., and L.T., Formal analysis, Ö.G. and K.G., Writing—Original Draft preparation, Ö.G., C.C., L.T., and K.G., Funding: C.C.

**Funding:** This research was funded by the University of Sydney's DVC Research Bridging Support Grant (G199771) and Cachet Group (G192167).

**Acknowledgments:** Authors would like to express their gratitude to all organizations and occupants for dedicating their time to participate in this study.

**Conflicts of Interest:** The authors declare no conflict of interest.

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
