# Peer review of "Differences in Occupants’ Satisfaction and Perceived Productivity in High- and Low-Performance Offices"

_buildings, doi:10.3390/buildings9090199_

Round 1
Reviewer 1 Report
Comm
This study analyzed the relationship between occupant satisfaction and perceived productivity in offices. It is a valuable contribution to the literature. My comments are as follows.
Line 30-34: I think all four key topics of investigation are some aspects of “IEQ”. The last bullet point should simply be “Office design” instead of “IEQ and office design”.
Line 35-45: This paragraph should include a brief mentioning of the importance of maintaining adequate ventilation for work performance. It’s a very relevant topic with plenty of objective data to support the relationship. Here are some examples:
Seppänen, O., W.J. Fisk, and Q.H. Lei, Ventilation and performance in office work. Indoor Air, 2006. Maddalena, R., et al., Effects of ventilation rate per person and per floor area on perceived air quality, sick building symptoms, and decision making. Indoor Air, 2015. Allen, J.G., et al., Associations of cognitive function scores with carbon dioxide, ventilation, and volatile organic compound exposures in office workers: a controlled exposure study of green and conventional office environments. Environmental Health Perspectives, 2016. Zhang, X., et al., Effects of exposure to carbon dioxide and bioeffluents on perceived air quality, self-assessed acute health symptoms and cognitive performance. Indoor Air, 2017. Shan, X., et al., Comparing mixing and displacement ventilation in tutorial rooms: Students' thermal comfort, sick building syndromes, and short-term performance. Building and Environment, 2016. Maula, H., et al., The effect of low ventilation rate with elevated bioeffluent concentration on work performance, perceived indoor air quality and health symptoms. Indoor Air, 2017.Line 82: I think authors should at least mention in the Introduction that there are other ways to objectively measure productivity. “Perceived productivity” has significant limitations that should be made clear to readers. I see that this point was very briefly addressed in the Conclusion: “Since there is not a universally accepted measurement of productivity, a self-assessed measured was used” (Line 359-360). I think the paragraph on study limitation (Line 358-361) should be moved to earlier parts of the paper, it should not appear in Conclusion only. I see in Materials/Methods that “perceived health” and “overall satisfaction with the work area / building” (Line 101-102) were also asked in the survey. Why were these other metrics not analyzed? Did the authors determine they were not relevant or is there follow-up work to study those metrics separately?
Line 144-145. Reading from Table 2, there seems to be a significant difference between 10-LPO and 10-HPO in the % of buildings with a specific office layout: “open-plan with low partitions 61%” is the dominant type for 10-LPO, while “open-plan without partitions 73%” is the dominant type for 10-HPO. Suggest authors to provide some highlight comparison re: how and if 10-LPO and/or 10-HPO were different from the 77-buiding dataset in any meaningful ways in terms of sample size, office layout, tenant type, building certification, and workplace arrangement.
Line 147-152. Does the data presented in Table 3 describing 10-LPO and 10-HPO alone, or is it referring to all 77 buildings? Similar to the above comment, I think it is informative for authors to comment if survey respondents from 10-LPO and 10-HPO differ in potentially meaningful ways compared to the 77-buildnig dataset.
Line 206-207. “Table 4 presents the R2s and standardized regression coefficients (b) of the two regression models.” Do authors mean Table 5 instead? I see 3 sets of regression results, not “two regression models”. What is the overall R2 for each regression model? Authors highlighted in Table 5 the two strongest predictors, but see that the 3th and 4th factors are not far behind. For example, it appears from Table 5 that “thermal comfort & air quality” is also important.
Figure 2 and 3 followed the discussion that Table 6 found “perceived productivity was mostly related to the occupants’ working experience and hours at that office, window proximity, office layout and workplace arrangement” (Line 226-228) specifically for 10-HPO. Are the results in Figure 2 and 3 also specific to 10-HPO? If yes, this needs to be made clearer.
Line 237-239. “When the working experience was taken into account, the group working for more than 5 years has been most adversely affected by the lack of outdoor environment connection.” But do authors have a plausible explanation for why lack of outdoor environment apparently only adversely affect those who worked 1-2 years, and >5 years, but actually benefit those working 2-5 years? Overall, I question if the results shown in Figure 2 and 3 are reliable enough to draw meaningful conclusion. For example, because the majority of data came from respondents working 30+ hours, I question if there’s sufficient data from respondents working fewer hours to support the statement that “Open-plan office with low partitions is the best type of office layout consider long working hours (Figure 3)” (Line 244-245).
Line 249-251, Line 254-256. Suggest authors making clear that those observations were limited to 10-HPO if that is in fact the case. It may not apply to other buildings.
Line 322-329. I think “thermal comfort & air quality” should also be mentioned, aside from individual space and personal control.
Line 333-335. Sentence is specific to 10-HPO only. This study did not provide data to generally support “workstation having open-plan office having low partition has best performance”. Sentence is misleading as written.
Line 355-357. Sentence is not supported by data provided. As I understand it, the analysis on “non-fixed workspace arrangement” is specific to 10-HPO only, and should not be implied to generally applicable to all office spaces.
ents attached.

Author Response
We would like to thank the reviewers for their constructive comments. In addition, a native speaker has edited the manuscript.
You may kindly find our revisions and respond to your comments below:
Reviewer 1:
This study analyzed the relationship between occupant satisfaction and perceived productivity in offices. It is a valuable contribution to the literature.
My comments are as follows.
Line 30-34: I think all four key topics of investigation are some aspects of “IEQ”. The last bullet point should simply be “Office design” instead of “IEQ and office design”.
Response: The last bullet point has been revised as “Office design”
Line 35-45: This paragraph should include a brief mentioning of the importance of maintaining adequate ventilation for work performance. It’s a very relevant topic with plenty of objective data to support the relationship. Here are some examples:
Seppänen, O., W.J. Fisk, and Q.H. Lei, Ventilation and performance in office work. Indoor Air, 2006. Maddalena, R., et al., Effects of ventilation rate per person and per floor area on perceived air quality, sick building symptoms, and decision making. Indoor Air, 2015. Allen, J.G., et al., Associations of cognitive function scores with carbon dioxide, ventilation, and volatile organic compound exposures in office workers: a controlled exposure study of green and conventional office environments. Environmental Health Perspectives, 2016. Zhang, X., et al., Effects of exposure to carbon dioxide and bioeffluents on perceived air quality, self-assessed acute health symptoms and cognitive performance. Indoor Air, 2017. Shan, X., et al., Comparing mixing and displacement ventilation in tutorial rooms: Students' thermal comfort, sick building syndromes, and short-term performance. Building and Environment, 2016. Maula, H., et al., The effect of low ventilation rate with elevated bioeffluent concentration on work performance, perceived indoor air quality and health symptoms. Indoor Air, 2017.Response: The relevant studies have been referred to. The paragraph has been revised as to include these studies and required explanations.
Line 82: I think authors should at least mention in the Introduction that there are other ways to objectively measure productivity. “Perceived productivity” has significant limitations that should be made clear to readers. I see that this point was very briefly addressed in the Conclusion: “Since there is not a universally accepted measurement of productivity, a self-assessed measured was used” (Line 359-360). I think the paragraph on study limitation (Line 358-361) should be moved to earlier parts of the paper, it should not appear in Conclusion only. I see in Materials/Methods that “perceived health” and “overall satisfaction with the work area/building” (Line 101-102) were also asked in the survey. Why were these other metrics not analyzed? Did the authors determine they were not relevant or is there follow-up work to study those metrics separately?
Response: The definition of perceived productivity has been mentioned in the Introduction and the limitations of the study have been emphasized again in Materials and Methods.
Results from overall satisfaction with work area were published by Buildings – Candido et al, 2019. In addition, these questions were the focus of previous and on-going work as they do deserve stand-alone analysis due to its significant effects on people. For this paper, we have chosen to use perceived productivity alone as a proxy for analysis.
Line 144-145. Reading from Table 2, there seems to be a significant difference between 10-LPO and 10-HPO in the % of buildings with a specific office layout: “open-plan with low partitions 61%” is the dominant type for 10-LPO, while “open-plan without partitions 73%” is the dominant type for 10-HPO. Suggest authors to provide some highlight comparison re how and if 10-LPO and/or 10-HPO were different from the 77-building dataset in any meaningful ways in terms of sample size, office layout, tenant type, building certification, and workplace arrangement.
Response: All the buildings have green certification and the tenant profile could be a topic of a follow-up study. However, further statistical analysis (two-way ANOVA) was done, in order to understand the impact of office layout and workplace arrangement.
Line 147-152. Does the data presented in Table 3 describing 10-LPO and 10-HPO alone, or is it referring to all 77 buildings? Similar to the above comment, I think it is informative for authors to comment if survey respondents from 10-LPO and 10-HPO differ in potentially meaningful ways compared to the 77-building dataset.
Response: Table 3 presents the occupant profile of 10-HPO and 10-LPO. The title of Table 3 has been revised to solve the confusion as “Summary of occupant profiles in 10-LPO and 10-HPO”. The occupant profile of all 77 buildings and focus groups (10-LPO and 10-HPO) were similar.
Line 206-207. “Table 4 presents the R2 s and standardized regression coefficients (b) of the two regression models.” Do authors mean Table 5 instead? I see 3 sets of regression results, not “two regression models”. What is the overall R2 for each regression model? Authors highlighted in Table 5 the two strongest predictors, but see that the 3th and 4th factors are not far behind. For example, it appears from Table 5 that “thermal comfort & air quality” is also important.
Response: There were typos, the number of the table and regression models have been corrected as “Table 5” and “3 regression models”.
Our study has shown the importance of the predictors Building/office aesthetics and quality, Personal control and Individual space. In order to emphasize this outcome, we focused only on these predictors as the two strongest predictors of each regression models.
Figure 2 and 3 followed the discussion that Table 6 found “perceived productivity was mostly related to the occupants’ working experience and hours at that office, window proximity, office layout, and workplace arrangement” (Line 226-228) specifically for 10-HPO. Are the results in Figure 2 and 3 also specific to 10-HPO? If yes, this needs to be made clearer.
Response: The further analysis have been done for each focus group, “however, no significant relationship was found for the focus group 10-LPO. On the other hand, significant results were found for the combination of some variables for the focus group 10-HPO”. The explanation in inverted commas has been given in the text before the Table 6.
Line 237-239. “When the working experience was taken into account, the group working for more than 5 years has been most adversely affected by the lack of outdoor environment connection.” But do authors have a plausible explanation for why lack of outdoor environment apparently only adversely affect those who worked 1-2 years, and >5 years, but actually benefit those working 2-5 years? Overall, I question if the results shown in Figure 2 and 3 are reliable enough to draw meaningful conclusion. For example, because the majority of data came from respondents working 30+ hours, I question if there’s sufficient data from respondents working fewer hours to support the statement that “Open-plan office with low partitions is the best type of office layout consider long working hours (Figure 3)” (Line 244-245).
Response: Each group has a normal distribution. However, for the office layout, we merged the group private offices and private offices shared with others, since the number of samples was small in that group. The pattern of the interrelations of the analyzed groups was similar but the new significant values were better. The figures and Table 6 have been revised according to the new results.
Line 249-251, Line 254-256. Suggest authors making clear that those observations were limited to 10-HPO if that is in fact the case. It may not apply to other buildings.
Response: The paragraph has been revised as “The study with the focus group indicates that open-plan office with low partitions is the best type of office layout considering long working hours (Figure 3)”.
Line 322-329. I think “thermal comfort & air quality” should also be mentioned, aside from individual space and personal control.
Response: “thermal comfort & air quality” has been mentioned for 10-HPO.
Line 333-335. Sentence is specific to 10-HPO only. This study did not provide data to generally support “workstation having open-plan office having low partition has best performance”. Sentence is misleading as written.
Response: The sentence has been revised as “The other determinant factor for perceived productivity was the height of partitions for 10-HPO, especially”. The following sentence has been revised as “When it is considered that 73.2% (Table 3) of employees are working long hours, it is evident that a workstation having an open-plan office with low partition has shown the best performance for 10-HPO”.
Line 355-357. Sentence is not supported by data provided. As I understand it, the analysis on “non-fixed workspace arrangement” is specific to 10-HPO only, and should not be implied to generally applicable to all office spaces.
Response: That sentence was from the paragraph, which summarizes the outcomes of our study. So at the beginning of the paragraph, this fact has been emphasized.
Reviewer 2 Report
The research on "Differences in occupants’ satisfaction and perceived productivity in high and low performance offices" is relevant in that it provides quantitative evidence of factors affecting performance in individual and open offices.
To the reviewer, the main contribution of this research is the differentiation and comparison in the analysis between high-performance and low-performance offices.
The paper is well written and organized, methodology is sound, and the analysis is sufficient. However, the findings are expected. The paper is acceptable for publication as is. However, aside from some minor typos, the reviewer recommends that the authors provide a better interpretation of the statistical results for readers not familiar with statistical analysis. For example, the text does not include any explanation of the meaning of "component loading"
Author Response
We would like to thank the reviewers for their constructive comments. In addition, a native speaker has edited the manuscript.
You may kindly find our revisions and respond to your comments below:
Reviewer 2:
The research on "Differences in occupants’ satisfaction and perceived productivity in high and low-performance offices" is relevant in that it provides quantitative evidence of factors affecting performance in individual and open offices.
To the reviewer, the main contribution of this research is the differentiation and comparison in the analysis between high-performance and low-performance offices.
The paper is well written and organized, methodology is sound, and the analysis is sufficient. However, the findings are expected. The paper is acceptable for publication as is. However, aside from some minor typos, the reviewer recommends that the authors provide a better interpretation of the statistical results for readers not familiar with statistical analysis. For example, the text does not include any explanation of the meaning of "component loading"
Response: Typos have been corrected. The term ‘component loading’ has been explained in Table 1 as “The contribution of each variable on principal components”.